# Understanding the Wellbeing Needs of First Nations Children in Out-of-Home Care in Australia: A Comprehensive Literature Review

**DOI:** 10.3390/ijerph21091208

**Published:** 2024-09-13

**Authors:** Darren Garvey, Ken Carter, Kate Anderson, Alana Gall, Kirsten Howard, Jemma Venables, Karen Healy, Lea Bill, Angeline Letendre, Michelle Dickson, Gail Garvey

**Affiliations:** 1School of Public Health, Faculty of Medicine, The University of Queensland, Brisbane, QLD 4006, Australia; g.garvey@uq.edu.au; 2Yardhura Walani, The Australian National University, Canberra, ACT 2601, Australia; kate.anderson@anu.edu.au; 3National Centre for Naturopathic Medicine, Southern Cross University, Lismore, NSW 2480, Australia; alana.gall@scu.edu.au; 4Menzies Centre for Health Policy and Economics, Faculty of Medicine and Health, The University of Sydney, Sydney, NSW 2050, Australia; 5School of Nursing, Midwifery and Social Work, The University of Queensland, Brisbane, QLD 4072, Australia; 6Alberta First Nations Information Governance Centre, Calgary, AB T2X 2A8, Canada; 7The Poche Centre for Indigenous Health, The University of Sydney, Sydney, NSW 2006, Australia

**Keywords:** out-of-home care, wellbeing, First Nations children, Australian

## Abstract

Introduction: Despite the increasing overrepresentation of Aboriginal and Torres Strait Islander (hereafter respectfully referred to as First Nations) children living in out-of-home care (OOHC) in Australia, little is known about their wellbeing needs. This comprehensive literature review aimed to identify these needs and the features of care required to meet them. Methods: MEDLINE, CINAHL, Scopus, Informit, PsycINFO, and Embase databases and relevant grey literature were searched from inception to December 2023 for articles presenting qualitative accounts and perspectives relevant to the wellbeing needs of First Nations children in OOHC. These included reports from First Nations children in OOHC; First Nations adults with lived experience of OOHC; carers, caseworkers, and organizational stakeholders; and First Nations community members with relevant lived and/or professional experience. We used the Preferred Reporting Items for Systematic Reviews and Meta-Analyses (PRISMA) guidelines and Braun and Clarke’s reflexive thematic analysis method for data analysis. Results: Thirty-five articles (19 peer-reviewed, 16 grey literature) met the inclusion criteria. Our analysis revealed six wellbeing needs of First Nations children in OOHC: *Being seen, being heard; a sense of stability; holistic health support; social and cultural connections; culturally safe OOHC providers; and preparedness for transitioning out of care*. A range of features of OOHC were also identified as critical for supporting these needs. Conclusions: Our findings suggest that First Nations children in OOHC have unique wellbeing needs in addition to safety, security, and health. Attention to the development and maintenance of social and cultural connections is an important concern that must be addressed by OOHC providers (caseworkers and organizations) and carers and supported by OOHC policy and the associated systems in Australia as part of providing culturally safe and supportive care.

## 1. Introduction

Out-of-home care (OOHC) refers to a range of alternative overnight accommodation options for children and youth who are deemed unable to live in their family home by a statutory authority due to child protection concerns [1]. OOHC arrangements can be broadly categorized into foster, kinship, or residential care; family group homes; or other living arrangements (e.g., private boarding) [2,3], and can be of short or longer-term duration. In Australia, a vastly disproportionate number of First Nations children are placed in OOHC compared with non-First Nations children. In 2022, First Nations children aged 0–17 years old comprised only 6% of this age group in the Australian population but represented 43% of children living in OOHC [4], with 45% of First Nations Australian children in OOHC having been continuously in care for over five years [5]. Calls to address this overrepresentation are numerous and longstanding.

First Nations organizations such as the Secretariat of National Aboriginal and Islander Child Care (SNAICC) have urged action to reduce the number of children requiring OOHC and improve the conditions of care. Recently, the National Framework for Protecting Children emphasized First Nations Australian children constitute a priority group and that reducing their overrepresentation in child protection must remain a focus [6]. The National Closing the Gap agreement is an agreement between Aboriginal and Torres Strait Islander peak organizations and all Australian governments and has the goal of reducing the overrepresentation of First Nations children in OOHC by 45% by 2031 [4]. However, forecasts project the number of First Nations children in OOHC will grow over the next decade—this raises important questions regarding the quality of the care provided for First Nations children and how it can best address their health and wellbeing needs.

For many First Nations people, the practice of removing children from family and Country (the land, waterways, and seas with which they are ancestrally and spiritually connected) carries additional meaning stemming from a period in Australia’s history where First Nations children were forcibly removed from their families through government policies. Some First Nations people may perceive parallels between present-day protection and past policies [3,7]. Within this broader historical context, present-day OOHC may be regarded warily, despite the child protection system having transitioned significantly from carrying out legislative acts of forced removal to prioritizing the protection and safety of children. However, acknowledging this historical legacy emphasizes the need to provide better care experiences for children that also address the concerns of First Nations families and communities.

First Nations conceptions of health and wellbeing diverge substantially from the dominant Western biomedical paradigm that perceives health as the absence of illness [8,9,10,11,12,13]. In the Australian context, many First Nations people understand their health and wellbeing as holistic and deeply interconnected with culture, Country, family, and community; identity and spirituality; and impacted by their work, education, basic needs, and physical and mental health [9,10,11]. For young First Nations Australians, wellbeing can also encompass striving to embody or maintain traditional ways within the context of intergenerational and cumulative trauma while navigating an uncertain future amidst dominant Western societal constructs, expectations, and values [12,14].

Central to First Nations’ concepts and experiences of wellbeing are the connections one has to a multifaceted social and spiritual heritage; the knowledge of these provides a basis for one’s development as a culturally grounded individual. This cultural development, based on the crucial roles that social, cultural, and spiritual connection play in a child’s life, is a priority for the health and wellbeing of Indigenous children in OOHC internationally. The United Nations Convention on the Rights of the Child [15] stipulates that children have the right to enjoy their culture, and the United Nations Declaration on the Rights of Indigenous People [16] also holds that Indigenous children have a fundamental right to learn about and sustain their cultural identities.

The degree to which cultural connection is supported and developed in OOHC is of particular concern. Of the 19,800 First Nations Australian children in OOHC in June 2023, only 32% were placed with their First Nations relatives or kin [5]. The high proportion of long-term placements puts these children at significant risk of long-term damage to the cultural and familial connections imperative for building and maintaining wellbeing [17]. In Australia, the Aboriginal and Torres Strait Islander Child Placement Principle (ATSICPP) stipulates that all First Nations children in OOHC should have a cultural support plan to ensure the preservation and enhancement of their cultural identity and prioritizes nurturing cultural connections via placing children with kin [18]. 

Given the importance of family and cultural connection to the wellbeing of First Nations youth [9], understanding the wellbeing needs of First Nations children in OOHC is critical. To the best of our knowledge, this article is the first comprehensive systematic review of the literature on the wellbeing needs of these children and the features of care required to meet them. While conceptualizations of wellbeing for First Nations peoples have been identified in broad terms, there is a paucity of literature focusing on the specific wellbeing needs of First Nations children in OOHC [19,20]. This review seeks to address this gap in order to provide an understanding of these needs, informed by First Nations children and those involved in their care. This is essential to informing child protection policy and practice reform so that First Nations children’s wellbeing can be addressed in culturally appropriate and meaningful ways.

## 2. Materials and Methods

This systematic review of peer-reviewed and grey literature was prepared and reported according to the Preferred Reporting Items of Systematic Reviews and Meta-Analyses (PRISMA) 2020 guidelines [21] and was registered with the Prospective Register of Systematic Reviews database (PROSPERO identification number: CRD42022375959). A strengths-based research approach and lens was applied across all stages in this review, as the focus was to identify the wellbeing needs of First Nations children in OOHC and the aspects of care to support them [22].

### 2.1. Research Team

The research team included First Nations peoples from Australia and Canada (DG, AG, LB, AL, MD, GG), qualitative researchers (KA, KHe, DG, GG, JV, AG), OOHC researchers (KHe, JV), researchers with experience in researching wellbeing in First Nations peoples (KA, GG, AG, AL, KHo), a health economist (KHo), and a research assistant (KC) working in First Nations health research. Members of the team also have lived experience in and with the OOHC system, providing personal insight and perspective to the review (AG, GG). We were conscious of the importance of our backgrounds and perspectives and reflexively considered how such factors may influence our approach to research [20,21].

### 2.2. Search Strategy

The MEDLINE, CINAHL, Scopus, Informit, PsycINFO, and Embase databases were searched for peer-reviewed literature published in English from inception to 31 December 2023.

Search terms were divided into three main categories: OOHC, wellbeing, and First Nations Australians. These were derived from published literature reviews focusing on at least one or a combination of OOHC [23,24,25], wellbeing [9,13], and First Nations Australian [9] research foci. Richardson et al. (2007) identified the importance of culture as a key determinant of the wellbeing of First Nations children in OOHC, so we added “cultur*” to the wellbeing search terms [20]. Table 1 outlines example search terms, and Appendix A details the complete database search strategy.

Title and abstract (TI/AB) field restrictions were utilized within the search strategy as a preliminary test of full texts, and TI/AB-restricted searches were conducted in all respective databases. Full-text search results did not provide any additional relevant papers. The reference lists of included peer-reviewed articles were also searched.

National and State government and non-governmental organization (NGO) websites and grey literature databases were examined for articles presenting accounts and perspectives relevant to the wellbeing of First Nations Australian children in OOHC. We used one or a combination of the following keywords: OOHC, First Nations Australians, wellbeing, and culture. Grey literature searches were conducted in Google Scholar, Australian Indigenous Health*Info*Net, and the SNAICC websites. Google Scholar searches were conducted using incognito or private browser mode, with the cookies and cache cleared before searching. Relevant articles were screened and selected using the same eligibility criteria applied to the peer-reviewed literature (see Appendix A).

### 2.3. Eligibility Criteria

Both peer-reviewed and grey literature articles reporting on the qualitative experiences of First Nations Australian children in OOHC were eligible for inclusion in this review if they reported direct perspectives of First Nations Australian children in OOHC, First Nations Australian adults with lived experience of OOHC, carers, caseworkers and organizational stakeholders, or First Nations Australian community members with lived and/or professional experience with OOHC. All included articles presented primary quotes or case study findings using qualitative data collection methods (focus groups, individual interviews, surveys, case studies, etc.). Articles were included from inception to the search data of 31 December 2023 to capture as many relevant articles as possible, due to the paucity of research in this area.

We excluded articles not written in English, with findings applying only to a particular disease or condition, comprising case studies or reports of neglect and abuse, or those with no qualitative data reported (i.e., quantitative studies), and trials, study protocols, theses, books or book chapters, media documents, and conference abstracts. Refer to Appendix A for the exclusion criteria hierarchy applied to screening both title/abstract and full-text articles, ranking the order of exclusion reasons which reviewers checked while screening articles.

### 2.4. Study Selection

The search results of the peer-reviewed articles were imported into EndNote 20 [26] referencing software, and duplicate articles were removed. The peer-reviewed articles were then imported from Endnote into Covidence [27] as an XML file. Duplicate articles for the grey literature search were identified and removed using Microsoft Excel (for Microsoft 365) [28]. AG and KC then performed independent TI/AB screens of approximately 10% of the peer-reviewed articles using the Covidence platform [27] and 10% of the grey literature results using Microsoft Excel (for Microsoft 365) [28]. The reviewers then met to review and resolve any conflicts in screening decisions by discussing their screening approach and thought process according to the eligibility criteria. Once all conflicts were resolved, KC completed the remainder of the TI/AB screening process. Full-text articles from both peer-reviewed and grey literature sources were then independently screened by KC and AG with support from a research assistant. KC searched the reference lists of the included peer-reviewed articles, with relevant articles screened independently by AG and KC for eligibility. Following the full-text article screening, the reviewers met to discuss and resolve any conflicts in inclusion/exclusion decisions according to the eligibility criteria. Figure 1 presents a PRISMA flow diagram detailing the articles identified, screened, excluded, and included in this review.

### 2.5. Data Collection and Analysis

Data was extracted using NVivo software [29]. KC and DG independently extracted the first five studies and compared their findings to ensure the consistency and reliability of all identified study characteristics. KC then completed the data extraction using the following headings: publication information (author(s); year published; article title; aim and location of the study; study methods; participant details (number of participants, age, and gender); participant perspective; and placement type (see Appendix A).

Guided by Braun and Clarke’s (2019) reflexive thematic analysis methods [30,31], KC and DG used an inductive analysis approach alongside a strengths-based lens [22,32]. Coding was stratified based on participant perspective (First Nations children/youth in OOHC and First Nations adults with lived experience in OOHC; carers; caseworkers and organizational stakeholders; and community). KC and DG independently coded nine studies and discussed the themes emerging across the participant perspectives, agreeing that they were sufficiently similar to warrant unstratified analysis of the data. KC and DG met three times and mind-mapped the emerging themes and subthemes while reflecting on the data. KC, DG, KA, and GG then met to revise and organize the preliminary themes and subthemes to iteratively develop and synthesize the ‘wellbeing needs’ (themes) and ‘supporting aspects of care’ (subthemes). AG, JV, KHe, AL, and LB provided feedback and suggestions to refine the findings further. Our reflexive process of data extraction and our open consultation and discussion incorporating the unique perspectives of the research team ensured a First Nations lens was applied to all data analysis.

## 3. Results

### 3.1. Paper Characteristics

Thirty-five articles (19 peer-reviewed, 16 grey literature) were included in the review. Most reported on the perspectives of multiple stakeholders. Nineteen studies explored the perspectives of First Nations Australian children in OOHC; 10 reported the perspectives of carers, 22 reported staff perspectives, and six reported community perspectives (see Appendix A).

All included articles reported primary qualitative data obtained using one or more methods; 11 via focus groups, 19 via interviews, four via one or more Aboriginal community-controlled organization (ACCO) consultation/submissions, and five via surveys. One study employed narrative art interviews, and another was conducted in the context of a youth forum. Two separate articles included the same participant population as other included articles in the review but presented additional qualitative findings of significance. Wellbeing was explored as a component of the broader research question in 33 articles; only two articles made wellbeing a focus of the main research aim. Thirteen studies reported both First Nations Australian and non-First Nations perspectives, and eight solely reported First Nations perspectives. Seven studies included the perspectives of staff and organizational stakeholders from ACCOs, NGOs, and government agencies but did not specify the individual backgrounds of participants, so it was unclear whether the sample included solely First Nations, or both First Nations and non-First Nations peoples.

### 3.2. Qualitative Results

Through our analysis of the articles, we identified six overarching wellbeing needs of First Nations Australian children in OOHC: *Being seen, being heard; a sense of stability; holistic health support; social and cultural connections; culturally safe OOHC providers; and preparedness for transitioning out of care* (refer to Table 2). These needs are explored below, including descriptions of aspects of care identified as being critical for supporting them.

### 3.3. Being Seen, Being Heard

Twenty-nine papers identified that the wellbeing of First Nations children and their experience of OOHC are strongly influenced by whether they feel seen and heard by the people making decisions about their OOHC circumstances [33,34,35,36,37,38,39,40,41,42,43,44,45,46,47,48,49,50,51,52,53,54,55,56,57,58,59,60,61,62]. When children feel seen and heard, they are imbued with a sense of autonomy and agency. Additionally, being treated by decision-makers as a person and not as a number is critical in supporting the wellbeing of children in care.

#### 3.3.1. Having Autonomy and Agency

Twenty-nine studies refer to acknowledging the autonomy and agency of children across the OOHC continuum, from entry into care to during care and transitioning out of care [33,34,35,36,37,38,39,40,41,42,43,44,45,46,47,48,49,50,51,52,53,54,55,56,57,58,59,60,61,62]. Each of these phases presents different challenges and opportunities to address children’s needs.

If children feel acknowledged and valued, they are more likely to open up and speak with their carers and caseworkers and feel empowered through their involvement in the decisions that affect them. Children who are consulted regarding their placement preferences and support needs upon entry into care feel valued by their carers and caseworkers.

*Talk to the actual kids, they’ll be very honest about their placement they don’t hide nothing get new workers to build trust with them kids and they will just spill their guts really*.(Sally, worker) [47]

Being seen and heard by decision-makers involves children being informed about their OOHC circumstances rather than excluded from decisions that impact on them. This makes them feel valued. A small gesture or explanation can do much to communicate that they matter.


*I think it’s like ya get dropped out of the loop… But basically we all are feeling so left out of things, these things are happening, we might be young but some sort of explanation would just go a mile…*
(Caroline, post-care, 19, Aboriginal) [33]

Good caseworkers establish trust and rapport with the child and their family through transparent consultations, enabling the child to safely voice their preferences and concerns across the continuum of care.

*In some cases, while FACS [Family and Community Services] involved the children in some decision-making, the reviewer identified that this consultation was not ongoing. Children have the right to be involved in decisions that affect them and impact their lives, and this failure to consult on an ongoing basis was identified as disempowering practice*.(Case file reviewer) [38]

#### 3.3.2. Being Treated Like a Child, Not a Number

The importance of recognizing the child in OOHC as a person, not just a statistic, was strongly emphasized in twenty-nine articles [33,34,35,36,37,38,39,40,41,42,43,44,45,46,47,48,49,50,51,52,53,54,55,56,57,58,59,60,61,62]. Ensuring that children in OOHC feel that they are cared for and matter as individuals is understandably a key aspect of supporting their wellbeing.

*It is so important to know the kids you are working with; each person is an individual. It is just bloody critical that these kids are seen, known and not just a number. This is the work I am so passionate about*. (Child Protection staff member) [33]

*They should listen to young people. Hear what they have to say to ask them what they think and if they are safe*. (15-year-old First Nations girl in care) [52]

Family/kin caregivers play a pivotal role in ensuring that children in OOHC feel seen, valued, and wanted.

*I was lucky ’cos my grandparents made me feel like I was part of the family. I was never, ever introduced as their foster child and that made me feel loved and appreciated. I would encourage foster carers to try and do that, that was the most important thing*. (Caden, post-care, 19, Aboriginal) [33]

First Nations caregivers are also regarded by children in care as providing critical connections to culture, family, and community.

*My foster carers…were Aboriginal. They taught me stuff about culture. They helped me keep in contact with family. I stuck with one for most of it… They listened to me… [Carer] was really understanding. He understood why I was misbehaving sometimes*. (Phoebe, returned home, 16, Aboriginal) [33]

### 3.4. A Sense of Stability

Twenty-nine papers recognized the importance of a sense of stability for the wellbeing of First Nations children in OOHC [33,34,35,36,37,38,39,40,41,42,43,45,46,47,48,49,50,51,55,56,57,58,59,60,61,62,63,64,65]. Placement stability refers to a sense of consistency and belonging felt in OOHC, characterized by the development and maintenance of key relationships with carers, caseworkers, friends, and family; continuity of attendance at the same school; and access to consistent support services.

#### 3.4.1. Experiencing Placement Stability

Given the sometimes-unstable home environments children have been living in, efforts to provide placement stability are critical, and the consequences of achieving this are profound. Twenty-one papers highlighted the need for experiencing supportive, stable placements to support wellbeing [33,34,35,36,38,40,41,42,43,44,45,49,50,51,54,55,58,60,61,63,65,66].

*Stability for Aboriginal people is grounded in their sense of identity in connection to family, kin, culture and country. In our view, permanent care/adoption potentially places an emphasis on achieving stability of living arrangements and a secure legal status potentially at the cost of the child’s identity and enduring relationships with their extended family and connection with community and culture*. (Victorian Aboriginal Child Care Agency) [34]

OOHC placements that facilitate and maintain connections with family, community, culture, and Country are the preference and priority for any permanent care options. Short-term stability is weighed against the potential damage to identity of a longer-term culturally unsafe placement.

#### 3.4.2. Receiving Support in School

Thirteen papers identified school as an essential element of wellbeing for First Nations children in OOHC [33,35,36,38,39,40,41,44,46,48,49,50,54,62]. Supportive school-based education can offer children in OOHC a sense of stability, facilitate social and intellectual development, and support future opportunities for employment and further education. Children who have positive experiences in school are often supported by caring teachers who help to ensure that they feel seen and heard.

*I always tell them [First Nations children in OOHC engaging in education] who I am and what I’m there to do. And then I ask them if they want to. Because I make sure that they are involved in the decision making of being involved. And I did have two kids go, ‘No. I’m not quite sure’. But then in the end, became involved. I think part of it is also listening to them about what they want*. (Education engagement intervention program teacher/mentor) [62]

#### 3.4.3. Being on a Pathway to Culturally Appropriate Permanency

Twenty-six papers referred to culturally appropriate permanency as a key aspect of care for First Nations Australian children in OOHC [33,34,35,36,37,38,40,41,42,43,45,46,47,49,50,51,55,56,57,58,59,60,61,62,63,64]. While permanency of care affords the potential for stability and security, it remains contentious due to the qualities associated with different types of permanency. For example, while adoption represents a permanent placement solution, it is not preferred or promoted by ACCOs, particularly if it is not guided by policy that prioritizes ongoing connections between children and their family, kin, community, culture, and Country:

*SNAICC submitted that permanency for Aboriginal children was ‘tied to existing identity, kinship relationships, and connections to culture and country’, and that it was important not to permanently deprive children of these connections through the application of ‘inflexible permanency planning measures*. (SNAICC) [38]

### 3.5. Holistic Health Support

The physical, emotional, and social health of children upon entry and throughout their time in OOHC are identified as critical for their wellbeing. The holistic health needs of each child are best understood and addressed if they are seen and heard as individuals and given opportunities to safely voice their concerns.

#### 3.5.1. Fulfilment of Basic Needs

Twenty-five articles reference the fulfillment of basic needs as foundational to the positive wellbeing of First Nations children in OOHC [2,33,34,35,36,37,38,39,40,41,42,44,45,46,47,48,49,50,51,52,54,56,58,60,61,62,67]. The adequate provision of the basics of life in OOHC, such as safety, food, and accommodation, is crucial for children to feel safe and protected.

*Most of us kids, the reason why we are in care is because our families are not reliable. You know, money problems, food, clothes, safety problems… The whole reason why they took us off our family was because we feel unsafe, we don’t feel much protected, there’s no food, and we’re not getting clothes… we’re not getting anything. But what’s the point of that if they do exactly the same in all these houses. It’s not better either way: living with our family, living with DCP [Department for Child Protection], government homes… or living on the streets… it’s not good anywhere*. (17-year-old Aboriginal male, residential care) [35]

Placements require ongoing monitoring for their capacity to provide basic and essential amenities foundational to placement stability and wellbeing.

#### 3.5.2. Receiving Care for Health and Physical Wellbeing

Ten articles identified access to timely medical assessments upon entry into care and regular medical appointments as essential to ensuring First Nations children are healthy [33,36,37,38,40,41,42,49,54,61]. Several reports identify children with severely delayed or unaddressed diagnoses of a range of debilitating conditions, including depression, post-traumatic stress disorder, tooth decay, and fetal alcohol spectrum disorders. The impact on the wellbeing of a child if such conditions remain unidentified and untreated is significant and unacceptable.

*I think another trend that we found is that we’ve got a number of young people who have gone through the care system to be diagnosed as foetal alcohol syndrome at 18. And they’ve already been in and out of detention and they’ve got involvement with the justice system, and now they’re 18, it’s the adult justice system, which is a real concern. One young fella in particular I’m thinking of, was actually in residential care and wasn’t diagnosed until he was 18*. (Western Australian NGO) [49]

#### 3.5.3. Provision of Trauma-Informed Care

Twenty-four papers referenced trauma-informed social and emotional wellbeing support for First Nations children in OOHC [33,34,35,36,37,38,39,40,41,42,43,44,45,47,48,49,50,51,52,53,54,57,61,65]. Proactive psychological and behavioral supports are essential in enabling children to process and heal from the various traumas, stressors, and challenges they experience leading up to and during placement in OOHC.

A child’s wellbeing before they enter care and their transition into care both greatly impact their wellbeing in OOHC. Entry into OOHC often occurs in traumatic or unstable circumstances, where children are vulnerable to physical or psychological harm or are dealing with trauma. First Nations children may enter OOHC with trauma associated with abuse, neglect, or harm.

*FACS fails to acknowledge that the removal of Aboriginal children from their families often exposes them to danger and ‘immense trauma’, as opposed to ‘protection’, (National Congress of Australia’s First Peoples) and that FACS intervention in and of itself is an extremely arduous, traumatic process that is actively harmful to all involved, particularly children*. (Grandmothers Against Removals New South Wales) [38]

*A lot of kids have had severe trauma, been too exposed to a lot of negative experiences, and you can see it, like behavioural change. A lot of the kids are getting suspended all the time, they’re acting out, they just show all the different traits, like physically, emotionally. You can see, spiritually, that they’re impacted too, on a lot of different levels. Their confidence is low, self-esteem, yeah, just a lot of different things*. (NSW ACCO) [49]

Behavioral and counseling supports were identified as factors facilitating the social and emotional wellbeing of First Nations children navigating their respective stressors, challenges, and trauma in OOHC. One article provided an example of a holistic and individualized approach to supporting children with a safe avenue through which to process their experiences and difficulties:

*While in placement, with the support of a strong and therapeutic care team, an appropriate cultural support plan and a KESO [Koorie Engagement Support Officer], Molly’s [Aboriginal girl in OOHC] behaviours have settled. Molly has told child protection she feels safe and secure with her carers*. (caseworker/reviewer) [42]

### 3.6. Social and Cultural Connections

Social and cultural connections are the relationships children have with family, kinship networks, community, Country, and culture. Thirty-five papers recognized the importance of maintaining and building these connections, which contribute to a strong sense of wellbeing by ensuring that children feel that they are an important part of a supportive network [33,34,35,36,37,38,39,40,41,42,43,44,45,46,47,48,49,50,51,52,53,54,55,56,57,58,59,60,61,62,63,64,65,66,68].

#### 3.6.1. Fostering Interconnected Relationships

Thirty-five papers identified the importance of interconnected relationships between First Nations children in OOHC, and their parents, siblings, kin, community, Country, and culture for wellbeing [33,34,35,36,37,38,39,40,41,42,43,44,45,46,47,48,49,50,51,52,53,54,55,56,57,58,59,60,61,62,63,64,65,66,68]. Multifaceted connections help mitigate feelings of isolation by instilling a sense of belonging and how the child fits into a community. Maintaining these connections is difficult when OOHC placements put the child at a distance from important people, sources of cultural information, and sites of connection.

*The major difficulty in the urban setting was appropriately placing children culturally, working out where they belonged*. (ACCO staff) [65]

#### 3.6.2. Maintaining Cultural Knowledge and Identity

Thirty-one studies referenced the development and maintenance of cultural knowledge and identity as an essential component of wellbeing for First Nations Australian children in OOHC [33,34,35,36,37,38,39,40,41,42,43,44,45,46,47,48,49,50,51,52,55,56,57,58,59,60,62,64,65,66,68]. The wellbeing of these children is supported by opportunities to develop cultural knowledge and learn about their place within it. A strengths-based focus of this information promotes a sense of self-esteem and pride in their First Nations heritage.

*Being Aboriginal is the proudest thing in my life, to know that that’s my people. It made me so proud to see what we’ve actually done and how far we’ve come to this day. It taught me that no matter what, I can still get up and do what I want*. (Aboriginal child in OOHC) [36]

Carers and role models provide cultural support and guidance, positively engaging with children to fortify their social networks. This enables children to see how they are part of a larger network comprising their family, kin, and broader community. Children can develop a sense of their place in the world instead of feeling isolated and alienated. OOHC placement with carers and caseworkers that facilitate meaningful cultural education and experiences supports the development of a strong and positive cultural identity. This instills a healthy and positive regard for First Nations culture, encouraging them to walk proudly as informed and connected First Nations people.

*They [First Nations children and youth upon entering cultural camps] didn’t know their connections to communities, didn’t know about the language, didn’t eat Aboriginal food, they knew nothing at all [of their culture]*. (Aboriginal education officer) [36]

#### 3.6.3. Feeling Connected to Community and Country

Maintaining connections with community and Country was identified as a significant component of wellbeing in twenty-seven studies [33,34,35,36,37,38,39,40,41,43,44,45,46,47,49,51,52,53,54,55,56,57,58,59,60,62,64,66]. Children develop their cultural knowledge through their connections with community, while their identity is supported by encouraging and fostering their knowledge and appreciation of their Country.

*Aboriginal children coming into care should be placed in their own country. Just because they’re Aboriginal, isn’t good enough. You need to be placed with people who know your identity*. (Non-First Nations carer) [54]

The value of connections to community and Country extends beyond their time in care. Social and cultural connections help develop resilience by providing children with a direction to head towards and a safe and secure place to return to.

*Participants identified a strong cultural identity and effective connection with community as a powerful source of resilience for Indigenous young people during and post transition from care*. (ACCO and Government OOHC workers) [50]

#### 3.6.4. Continued Links to Family and Kin

Thirty-two articles acknowledged the value for First Nations Australian children in OOHC of being connected with their family and kin [33,34,35,36,37,38,39,40,41,42,43,44,45,46,47,48,49,51,52,53,54,55,56,57,58,59,60,61,63,64,65,66,68]. Consistent care provided by family members is regarded as the most preferable and culturally appropriate OOHC option for First Nations children. Ensuring children are connected to extended family and kin promotes a sense of familiarity and belonging and provides them with a supportive network. Kinship networks enable connection to culture and community and offer some children a comforting and familiar arrangement in otherwise disruptive and distressing circumstances.

*Give Aboriginal kids back to their home, their family, after you’ve gone through and made sure everything is all safe and all good. If not the mother and father, then maybe the kid has sisters, aunties, or an Aboriginal carer is available*. (Aboriginal caregiver) [36]

*Living there [in kinship care] feels like a family*. (Shane, kinship care, 15, Aboriginal) [33]

Kinship care is emphasized as the most appropriate placement type for First Nations children due to their familiarity with kin; this supports and maintains their sense of identity and other important connections.

*Well, the strength [of kinship care] is that children remain within their extended family, which supports our philosophy around self-determination, self-management. The family best knows the family circumstances*. (Jenny, worker) [37]

Sibling contact is a key component of wellbeing, especially when siblings are placed together in OOHC or remain within close proximity to maintain regular contact. Many children in OOHC prefer to be close to their siblings.

*His [First Nations, 8 year old boy in relative care] older siblings were scattered geographically but it was clear from his narrative that he wanted regular contact with his older siblings*. (OOHC team leader and art therapist) [52]

Knowing that siblings are close and accessible has a settling influence on a child in OOHC in circumstances otherwise characterized by uncertainty and upheaval; they highly value the presence of someone familiar and trustworthy.

*If I need to talk to someone now, my brother would be the first person I would talk to*. (Ellie, residential care, 16, Aboriginal) [33]

Family reunification was identified as a key priority in supporting the wellbeing of First Nations children in OOHC. First Nations carers and caseworkers from ACCOs develop and adhere to appropriate cultural support plans aiming to maintain children’s connections with family and kin throughout their OOHC experience.

*I had someone sit down with me and go through everything, my mob, my family. There is nothing else I need to know*. (Female, First Nations, 17 years) [48]

*I want to find out if I have a cultural support plan so I can get help finding more info about my culture and where my family was from*. (Female, First Nations, 14 years) [48]

Reunification with family is a primary and driving wish of many children in OOHC. For many, their distance from family is deeply upsetting, and the prospect of reunification provides hope and supports their wellbeing.

*Few months ago I asked [Department of Health and Human Services, Victoria] if I could find my dad. Haven’t seen him since I was one. Part of my life I’ve never met, so not good. My dad is the only actual family I know*. (Evan, foster care, 15, Aboriginal) [33]

#### 3.6.5. Being Supported by Friends

Nine papers emphasized the importance of friends for the wellbeing of First Nations Australian children in OOHC [33,35,36,40,46,51,56,59,62]. Friends can offer children in OOHC someone to confide in and be supported by as they navigate life in care. Connection with friends provides a unique avenue for emotional and social support, fun and entertainment, and a sense of belonging.


*Q: Who do you go to for support?*
*My friends, but more like my best friends. I’ve known them since I was like three and we’ve always stayed in contact and if I have a problem on my mind, I can always just go to his house*. (Ethan, kinship care, 15, Aboriginal) [33]

### 3.7. Culturally Safe OOHC Providers

Thirty-three papers identified the cultural safety and quality of care of OOHC providers as important components of wellbeing for First Nations children in care [33,34,35,36,37,38,39,40,41,42,43,44,45,47,48,49,50,51,52,53,54,55,56,57,58,59,60,61,62,63,64,65,66,68]. ACCOs were identified as being able to most appropriately facilitate wellbeing due to delivering services in a culturally safe manner.

One prominent example of an Aboriginal-led community care model are the Tangentyere Council Aboriginal Corporation OOHC houses, operating two OOHC houses in Alice Springs providing temporary care to children (0–12 years old) while their family are unable to [55]. Some of their key principles include having the children at the center of all decision making, having local community members caring for local kids, creating a sense of home with familiar routine, structures, and consistency, and connecting families to support [55]. Positive outcomes from these community-led OOHC houses include better care experiences for children, high staff retention, and long-lasting relationships with children and families who have lived in the houses [55].

#### 3.7.1. Supported by OOHC Organizations Trusted by First Nations Peoples

Certain OOHC agencies are preferred over others, particularly those that are viewed as having credibility with First Nations peoples. ACCOs were seen as having an affinity with and supporting the connections that are foundational to the cultural knowledge and identity of children in care.

*We understand where people [Aboriginal families] come from you can’t just have a mainstream organisation culturally competent, its philosophy is driven by white people, how they were raised, how they understand programs and services*. (ACCO staff) [47]

#### 3.7.2. Provision of Support Services Grounded in Culturally Safe Approaches

The extent to which organizations and staff demonstrate cultural safety impacts children’s wellbeing and their experience of OOHC. OOHC policy and staff that facilitate transparent and respectful engagement between children and carers underpin positive relationships that allow children a degree of autonomy and influence within their OOHC. Consequently, First Nations children can experience OOHC as a receptive environment in which they matter and where their immediate and ongoing needs are supported.

*Aboriginal community-controlled agencies are best placed to support Aboriginal children and young people in OOHC, including maintaining their connection to family, community, culture and Country that is central to identity development and wellbeing*. (New South Wales Council of Social Service) [38]

### 3.8. Preparedness for Transitioning Out of Care

Just as the transition into care poses particular challenges for the wellbeing of First Nations children, so too does transitioning out of OOHC. Wellbeing is supported when children have been prepared for changes associated with OOHC. When carers, caseworkers, and OOHC systems fail to provide effective planning, preparation, and focus on family reunification for life after OOHC, the transition out of care can leave many First Nations children vulnerable to revictimization. Eighteen papers identified preparedness for change as a key component of wellbeing [33,34,36,38,39,40,41,42,45,47,48,49,50,51,54,56,58,62].

#### 3.8.1. Given Adequate Opportunities for Reunification with Family

Valuing and respecting the needs and preferences of First Nations children extends to how their transition out of care is planned for and facilitated. Some children felt prepared and informed, while others described an abrupt, unsupported exit experience. However, children who were unsupported by the system were left feeling neglected and frustrated:

*He [14-year-old Aboriginal/South Sea Islander boy in OOHC] showed little attachment to the carer in that he talked of running away and not needing anyone*. (OOHC team leader and art therapist) [52]

Some took matters into their own hands by leaving care and self-placing before turning 18:

*We’ve got lots of kids walking from care and leaving at 15. And particularly going back to Country or trying to find Country*. (New South Wales NGO) [49]

#### 3.8.2. Provided with Life Skills for after Care

Preparedness for transitioning out of care involved developing essential independent living skills; education; maintaining connections to community, family, and culture; and having an appropriate transition care plan in place.

*We know there are 16, 17, 18-year-olds out there that can’t even boil water, you know, yet they want to fall pregnant; so if you can get it in there early enough to get these old people to teach these children survival skills, and not just Indigenous (skills), but also how to cook a meal and sew a button on*. (Carer) [39]

## 4. Discussion

This comprehensive literature review aimed to identify the wellbeing needs of First Nations Australian children in OOHC by analyzing their perspectives and those of caregivers, caseworkers, and stakeholders involved in decision-making and care provision. Six interconnected wellbeing needs were identified: *Being seen, being heard; a sense of stability; holistic health support; social and cultural connections; culturally safe OOHC providers; and preparedness for transitioning out of care.* Across these areas of need, corresponding features of OOHC were identified as critical for supporting the wellbeing of children in care over the duration of their stay and in preparation for their departure. The review indicates that the wellbeing needs of First Nations children in OOHC require attention from their earliest encounter with the system to ensure that they are humanized and heard. It is important to acknowledge that each child’s circumstance on entering OOHC is unique; this enhances the likelihood that a child’s basic and specific wellbeing needs are met.

There are similarities between First Nations and non-First Nations children’s wellbeing needs in OOHC. These include having their decision-making and autonomy acknowledged and respected, stability and preparedness to transition out of care, maintaining family contact, a preference for kinship care, addressing trauma, and having foundational needs met [33,48]. However, there are also notable differences; of significance is the importance of connections to culture and identity for First Nations children [33]. While attending to the key components of wellbeing underpinned by safe policies and attentive personnel is key for the wellbeing of all children in care, our review identifies needs requiring specific attention for First Nations children. Cultural connections were identified as an integral component of wellbeing for First Nations children; their maintenance and development throughout OOHC were important during care and beneficial when exiting care.

The absence of culturally responsive approaches to supporting First Nations Australian children, young people, and their parents and kin prior to and during their journeys through OOHC is a contributor to First Nations children’s over-representation in the child protection system and to the poor conditions and outcomes [69]. It was evident that cultural identity and connections are essential components of wellbeing, grounding First Nations children with a collective sense of belonging, meaning, and resilience. Being fortified in cultural knowledge and identity went hand in hand with being supported by a network of connections to family, kin, community, and Country [12]. Knowing who they are plays an important role in children’s navigation of life post-OOHC.

Trauma-informed social and emotional support was another reported wellbeing need of First Nations children. The review highlights the importance of trauma-informed approaches that prioritize consideration of the intergenerational trauma and historic and ongoing legacy of Stolen Generations. In their research on approaches to preventing the removal of First Nations infants, Chamberlain et al. (2022, p. 263) stated that “We must maximise therapeutic outcomes and promote therapeutic, evidence-based, community-led, culturally responsive, trauma-integrated interventions and practices” [69]. Our review indicates that these principles are critical to improving culturally responsive practices for First Nations children and families at all points of involvement in the child protection system, including their journeys through OOHC. However, children and others reported inadequate and inappropriate cultural support with poorly executed cultural support plans, a failure to recognize and respond to Indigeneity, and low proportions of children being case-managed by ACCOs over government agencies [33].

The need for connection with cultural knowledge and identity was reported as urgent and sensitive, given the genocidal history of the Stolen Generations and the fact that assimilatory policies and practices invoke negative memories and intergenerational trauma [70]. Contrary to current government efforts that push for permanency planning in the form of adoptions, policy and practice reform must shift towards prioritizing culturally appropriate permanency [55]. This would involve increasing support for kinship care and family reunification efforts to create conditions in which First Nations children and youth can develop and maintain connections to culture, family, kin, community, and Country.

Due to the unique experience of OOHC, children included who were no longer under the responsibility of child protection upon turning 18 years old were often unprepared in terms of independent living skills, educational progression, and connections to family, kin, community, culture, and Country. This is an important consideration for policy and practice reform, as the current OOHC environment may inadvertently see First Nations children leave the relative safety and stability of OOHC with few skills and fewer prospects [70]. A broader vision of OOHC is required, in which it is an experience that aims to support First Nations children in building their readiness for future opportunities through supported transitions out of care. Importantly, all Australian jurisdictions have recently announced and commenced the delivery of extended care programs offering financial and/or practical support to care leavers up to the age of 21 [71]. It is essential that the unique wellbeing needs of First Nations Australian care leavers are considered in the design and delivery of these programs to ensure they are culturally responsive and sensitive to the general and specific wellbeing needs of First Nations people.

Overall, the identified wellbeing needs and supporting aspects of care informed by this review offer guidance for OOHC providers and administrators in the evaluation of existing and future policy, programs, and practice. This would ensure that the wellbeing of First Nations children in OOHC is holistically supported, appropriately resourced, and systematically coordinated.

### Strengths and Limitations

This is the first comprehensive review to systematically report on the wellbeing needs of First Nations Australian children in OOHC and the aspects of care that support them. The perspectives of First Nations children were given prominence in this endeavor, offering first-hand insights into their OOHC experiences. This review thus adopted a strengths-based approach to considering the ways in which wellbeing needs can be positively framed to be addressed in the future. The other key strengths of this article were the unique and diverse positionality, experience, and reflexive lens of the research team and our comprehensive review methodology, which included key ACCO-led reports that would otherwise be excluded if the review only focused on peer-reviewed literature.

The intention was to privilege the voices of First Nations children in OOHC; however, our literature review confirmed the paucity of such work. Due to a lack of studies reporting solely on the perspectives of First Nations children in OOHC, the perspectives of caregivers, other stakeholders, community members, and OOHC staff were also included to inform the understanding of wellbeing needs of children. While the emerging themes and subthemes during data extraction highlighted significant similarities across wellbeing needs for First Nations children in OOHC, the inclusion of other participant perspectives risks the identified needs not being directly representative of the lived experience and views of First Nations children in OOHC.

## 5. Conclusions

This review is a timely contribution to the broader wellbeing project aimed at understanding and addressing the wellbeing needs of First Nations Australian people and Indigenous peoples internationally, grounded in their perspectives and based on their priorities. It supports ongoing efforts to improve OOHC for First Nations children and helps clarify elements that promote their wellbeing based on their experiences in care and the perspectives of those involved in the provision of care. While the onus for improving the quality of OOHC rests with individual care providers, the central roles that care organizations and appropriate policy play in outlining the standards and expectations of culturally safe care should not be underestimated. It will take a coordinated and committed effort to ensure that the wellbeing needs of First Nations children in OOHC are acknowledged and addressed in ways that meet their basic needs and provide them with opportunities to maintain important social and cultural connections. This includes attention to life skills, social skills, and cultural skills and knowledge that will, in turn, equip them to navigate life after OOHC. Care systems must be reoriented to better acknowledge the wellbeing needs of First Nations children and increase their adherence to principles and practices identified as supporting the cultural identity of First Nations children in OOHC. The care and support of First Nations Australian children must remain a key focus of OOHC services, informed by those directly impacted and involved.

## Figures and Tables

**Figure 1 ijerph-21-01208-f001:**
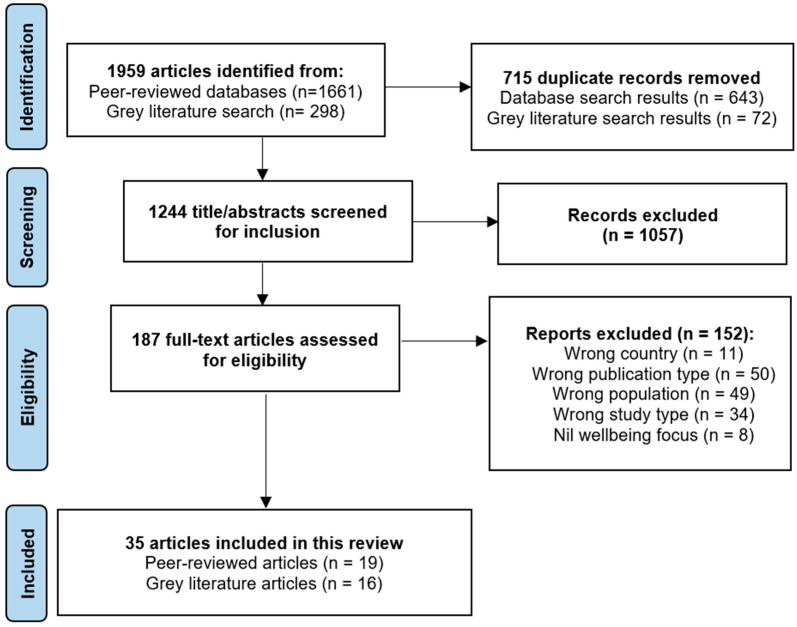
PRISMA flow diagram.

**Table 1 ijerph-21-01208-t001:** Example search strategy terms from Embase search.

	Subject Term	Search Terms
1	OOHC [22,23,24]	“out of home care” OR “out-of-home care” OR OOHC OR “out of home placement” OR “out-of-home placement” OR “residential care” OR “state care” OR “public care” OR “kinship care” OR “in care” OR “foster care” OR “foster family care” OR “foster home care” OR “foster child*” OR “guardian*” (TI/AB)
2	Wellbeing [9,13]	wellbeing OR well-being OR SEWB OR “quality of life” OR HR-QOL OR HRQOL OR QOL OR wellness OR “life quality” OR “health related quality of life” OR “health-related quality of life” OR “cultur*” (TI/AB)
3	First Nations Australians [9]	Aborigin* OR Indigenous OR “Torres Strait” OR “First Nation*” OR “First Australia*” (TI/AB)
4		1 AND 2 AND 3

Note: the truncation symbol * is used to inform the relevant database to find all variants of a given word following the symbol. For example, the term cultur* would also include searches for culture, cultures, cultured, cultural, and culturally.

**Table 2 ijerph-21-01208-t002:** Summary of wellbeing needs (themes), aspects of care (subthemes), and supporting quotes.

Wellbeing Needs (Themes) and Aspects of Care (Subthemes)	Quotes
**3.3. Being seen, being heard** 3.3.1 Having autonomy and agency 3.3.2 Being treated like a child, not a number	*I think it’s like ya get dropped out of the loop… But basically we all are feeling so left out of things, these things are happening, we might be young but some sort of explanation would just go a mile…* (Caroline, post-care, 19, Aboriginal) [33]
**3.4. A sense of stability** 3.4.1. Experiencing placement stability 3.4.2. Receiving support in school 3.4.3. Being on a pathway to culturally appropriate permanency	*Stability for Aboriginal people is grounded in their sense of identity in connection to family, kin, culture and country. In our view, permanent care/adoption potentially places an emphasis on achieving stability of living arrangements and a secure legal status potentially at the cost of the child’s identity and enduring relationships with their extended family and connection with community and culture.* (Victorian Aboriginal Child Care Agency) [34]
**3.5. Holistic health support** 3.5.1. Fulfilment of basic needs 3.5.2. Receiving care for health and physical wellbeing 3.5.3. Provision of trauma-informed care	*Most of us kids, the reason why we are in care is because our families are not reliable. You know, money problems, food, clothes, safety problems… The whole reason why they took us off our family was because we feel unsafe, we don’t feel much protected, there’s no food, and we’re not getting clothes… we’re not getting anything. But what’s the point of that if they do exactly the same in all these houses. It’s not better either way: living with our family, living with DCP [Department for Child Protection], government homes… or living on the streets… it’s not good anywhere.* (17-year-old Aboriginal male, residential care) [35]
**3.6. Social and cultural connections** 3.6.1. Fostering interconnected relationships 3.6.2. Maintaining cultural knowledge and identity 3.6.3. Feeling connected to community and Country 3.6.4. Continued links to family and kin 3.6.5. Being supported by friends	*Being Aboriginal is the proudest thing in my life, to know that that’s my people. It made me so proud to see what we’ve actually done and how far we’ve come to this day. It taught me that no matter what, I can still get up and do what I want.* (Aboriginal child in OOHC) [36] *Well, the strength [of kinship care] is that children remain within their extended family, which supports our philosophy around self-determination, self-management. The family best knows the family circumstances.* (Jenny, worker) [37]
**3.7. Culturally safe OOHC providers** 3.7.1. Supported by OOHC organizations trusted by First Nations peoples 3.7.2. Provision of support services grounded in culturally safe approaches	*Aboriginal community-controlled agencies are best placed to support Aboriginal children and young people in OOHC, including maintaining their connection to family, community, culture and Country that is central to identity development and wellbeing.* (New South Wales Council of Social Service) [38]
**3.8. Preparedness for transitioning out of care** 3.8.1. Given adequate opportunities for reunification with family 3.8.2. Provided with life skills for after care	*We know there are 16, 17, 18-year-olds out there that can’t even boil water, you know, yet they want to fall pregnant; so if you can get it in there early enough to get these old people to teach these children survival skills, and not just Indigenous (skills), but also how to cook a meal and sew a button on.* (Carer) [39]

## Data Availability

The data presented in this study are available online as listed in the references section.

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
