# Peer review of "Understanding the Wellbeing Needs of First Nations Children in Out-of-Home Care in Australia: A Comprehensive Literature Review"

_ijerph, 2024, doi:10.3390/ijerph21091208_

Round 1
Reviewer 1 Report
Comments and Suggestions for Authors
Thank you for the opportunity to critique this comprehensive literature review regarding the well- being needs of First Nations children in out of home care in Australia. This is a highly topical and important review given the excess numbers of Aboriginal children in out of home care and the need to understand their needs so that care can be re orientated to meet them.
The key strength of this study was the use of Aboriginal authorship, Aboriginal design and including first hand reports from First Nations children in out of home care as well as other important people such as caseworkers and organisational stakeholders.
Overall the literature review is excellently presented with concise, engaging and readable language and I commend this review for publication. I have highlighted a few reasonably minor issues for consideration.
Introduction
Generally very well written. An international audience may need more specific detail about the Closing the Gap strategy as well as a little more information about traumatic forced separations and permanency planning, which may not be familiar.
The rationale for undertaking this literature review is presented in lines 89 to 101 but I wonder whether this would fit best just before the materials and methods and could be integrated into the last paragraph from line 144 to 153, with a decrease in repetition.
Materials and methods
These are well described and it appears that the authors constitute an impressive health team. The authors did mention a health economist was involved but I don't believe there were any health economic analyses or comments through the article. This could be included if appropriate.
The search strategy is clear and well articulated and the data collection and analysis appears thorough.
Results
The general structure of discussing the theme and subtheme and then providing a quote worked well. Whilst very interesting, it took me a while to appreciate the overall themes had related subheadings. A table highlighting which are themes and subheadings would be helpful eg 3.1 with subheading 3.3.1. , 3.3.2 in a Table so one can see at a glance how it fits together. There is also a bit of repetition between the subthemes so could possibly condense into fewer subthemes
The order of presentation of themes is not clear. Was this from most to least common? Perhaps it would be more intuitive to go from more basic survival needs to higher level needs.
Discussion
This was again well presented, concise and thoughtful. My only comment would be line 676 where the authors mention that the absence of culturally responsive approaches are a major contributor to First Nations children's over representation in the child protection system. I would argue there are many other contributors and I don't there is evidence to suggest this in itself is a major one over and above the many adverse childhood experiences that have contributed. I would suggest leaving out the word ‘major’ in this context
Conclusions
These are appropriate and draw from the excellent body of work that has been presented. Thanks again for the opportunity to review
Author Response
Please see the attachment. There is a 'Tracked changes' version of the updated review manuscript resubmission. Thank you for your comments.

Reviewer 2 Report
Comments and Suggestions for Authors
We thank the authors for highlighting such an interesting and important topic. Kindly see below my comments:
1. Line 124 to 144 in the introduction needs to be deleted. At this point, the introduction is too long, and the supporting evidence provided for cultural and family connection in relation to youth well-being is sufficient.
2. What is missing in this manuscript are the tables. Please include a table with general study characteristics and a table highlighting the major themes/sub-themes and all relevant quotes. The text would reflect those themes and example quotes.
3. How did you assess the quality of the collected evidence? It would be useful to add a quality assessment tool such as the CASP checklist to look at how the evidence was qualitatively measured across included studies, particularly that this paper is a systematic rather than a scoping review.
Author Response

(The authors gave the same response as above.)

Reviewer 3 Report
Comments and Suggestions for Authors
Thank you for your work in this area. A comprehensive systematic search of the literature on the wellbeing needs of First Nations children in out-of-home care was conducted. The review considers an important public health challenge relevant to the journal's scope with implications for Indigenous peoples globally. There are two major strengths of the systematic review: (1) the lens applied to the search and analysis as informed by the diverse range of expertise, experiences, and knowledge sets of the research team; and (2) the inclusion of peer-review and grey literature capturing perspectives from a range of relevant stakeholders involved in the provision and delivery of out-of-home care. The manuscript is well-written but would benefit from a few revisions to strengthen the methodological approach described and the implications of the scoping review findings.
Introduction:
This section clearly describes the topic area's importance, the gap in knowledge, the work's contributions, and the objectives of the review.
Methods:
2.2. Search Strategy
· The search strategy described in A1 includes restriction of articles based in Australia and New Zealand. The authors could include a few more details to clarify why geographic location was restricted to also include New Zealand.
· This section can also be strengthened by describing the process that was established to resolve conflicts between reviewers during the title/abstract screen and full-text review. Was interrater reliability assessed? More information on the search strategy should be included in the protocol. Will this be provided as supplementary material?
· The section could also briefly describe how the search strategy was developed, including whether a research librarian was engaged in the process.
· It is a strength that the strategy included a systematic search of grey literature. The authors could specify how many pages in Google were reviewed (e.g., the first 10 pages).
· In what format were articles exported from databases to upload to Covidence?
2.3. Eligibility Criteria
· The authors may consider including a table outlining the eligibility criteria used to screen articles. This should be comprehensive, outlining key criteria components that informed the search strategy with a brief description of inclusion and exclusion (i.e., population, out-of-home care, geographic context, type of source, language, date, etc.). This section typically includes a justification statement for the date and geographic context.
Results:
3.6.1 Feeling connected – could be more descriptive by rephrasing as “fostering interconnected relationships.” The authors may have a better suited word to describe the subtheme under social and cultural connections.
3.7. Culturally Safe OOHC Providers
· The authors state, “Thirty-three papers identify the cultural safety and quality of care of 588 OOHC providers as important components of wellbeing for First Nations children in care [34–44,46–67,69].” The results could be strengthened by incorporating a few examples of culturally safe practices that were applied by OOHC providers. This would be of tremendous value to readers looking to adopt culturally safe approaches to better respond to the wellbeing needs of First Nations children in OOHC.
Discussion:
· The discussion could be strengthened by highlighting practice implications, including how the information presented in the paper can be used by providers and/or health systems administrators involved in program planning to provide culturally safe and supportive care.
· Under strengths and limitations, the application of a strengths-based approach is mentioned. I agree that a strengths-based approach was applied; however, this should be noted initially in the methods section and how the analysis was approached.
Author Response

(The authors gave the same response as above.)

Round 2
Reviewer 2 Report
Comments and Suggestions for Authors
We thank the authors for addressing all feedback and concerns. No additional changes are needed.